# The Role of Hypoxia-Associated Long Non-Coding RNAs in Breast Cancer

**DOI:** 10.3390/cells11101679

**Published:** 2022-05-18

**Authors:** Vilma Maldonado, Jorge Melendez-Zajgla

**Affiliations:** 1Cancer Functional Genomics Laboratory, Instituto Nacional de Medicina Genomica, Periférico Sur No. 4809, Tlalpan, Mexico City 14610, Mexico; vmaldonado@inmegen.gob.mx; 2Epigenetics Laboratory, Instituto Nacional de Medicina Genomica, Periférico Sur No. 4809, Tlalpan, Mexico City 14610, Mexico

**Keywords:** long non-coding RNAs, hypoxia, breast cancer, lncRNAs

## Abstract

Breast cancer is the leading cause of cancer-related deaths in women worldwide. In the United States, even with earlier diagnosis and treatment improvements, the decline in mortality has stagnated in recent years. More research is needed to provide better diagnostic, prognostic, and therapeutic tools for these patients. Long non-coding RNAs are newly described molecules that have extensive roles in breast cancer. Emerging reports have shown that there is a strong link between these RNAs and the hypoxic response of breast cancer cells, which may be an important factor for enhanced tumoral progression. In this review, we summarize the role of hypoxia-associated lncRNAs in the classic cancer hallmarks, describing their effects on the upstream and downstream hypoxia signaling pathway and the use of them as diagnostic and prognostic tools.

## 1. Introduction

Breast cancer is the leading cause of cancer-related deaths in women worldwide. Its incidence has been increasing 0.5% per year since the last decade, mainly due to the decline in fertility rate and the increase in body weight among young women [1]. In the United States, even with earlier diagnosis and treatment improvements, the decline in mortality has stagnated in recent years, from a reported 3% to a 1% annually [2], so we need renovated research efforts to address this.

One of the most important mechanisms that drive breast cancer progression is hypoxia—a decrease in the microenvironment oxygen tension. Hypoxia in tumors results from the growth of aberrant new blood vessels developed during cancer progression, which cannot sustain an adequate blood supply. Tumor cells located more than 180 µm away from a vessel become anoxic and die by necrosis [3]. Cells in the immediate vicinity can survive in a chronic hypoxic condition by eliciting a strong cellular hypoxic response [4]. This response induces several local and distant physiological mechanisms, including a shift from aerobic to anaerobic cellular respiration, production of growth factors, pH regulation, proliferation, induction of distant production of erythropoietin from the kidney and local neo-angiogenesis.

## 2. Hypoxia Signaling Pathway

### 2.1. Canonical Hypoxia Signaling Pathway

Cells respond to hypoxia by a concerted physiological and sometimes pathological responses by activating the Hypoxia Signaling Pathway (HSP), which is centrally guided by a group of related proteins termed Hypoxia-Inducible-Factors (HIF). The HIF family is a conserved group of transcription factors that act as a heterodimer of alpha and beta subunits. In humans there are three alpha (HIF-1α, HIF-2α/EPAS and HIF-3α) and two beta paralogs (ARNT, ARNT2) [5]. In normoxic conditions, the canonical HIFα subunit, HIF-1α is bound to the von Hippel-Lindau (VHL) protein, allowing the activation of the ubiquitin ligase system, which renders it susceptible to its degradation by the proteasomal degradation complex (Figure 1). To bind VHL, HIF-1α proline residues need to be hydroxylated, a process that depends on several hydroxylases, including α-ketoglutarate-dependent dioxygenases and the prolyl hydroxylases (PHD). In addition, hydroxylation of an asparagine in the C-terminal transactivation domain by the asparaginyl hydroxylase (factor-inhibiting HIF (FIH)) prevents its interaction with the p300 coactivator and thus HIF transcriptional activity [6,7]. An oxygen level decrease inhibits the PHD and FIH, leading to a reduction in hydroxylation and thus, HIF-1α stabilization. Higher levels of this unit allow it to dimerize with the HIF-1β subunit, which induces their translocation to the nucleus. There, the dimer recruits additional co-activators such as CREB and p300 and acts as a transcription factor that binds to E-box-like hypoxia response elements (HREs) (5′-RCGTG-3′) in a diverse array of hypoxia-inducible promoters in at least a couple hundred genes, as described by integrative approaches [8]. The products of these genes not only regulate various biological processes, including cellular metabolism, growth, apoptosis, and migration, but also include several oncogenes and tumor suppressor genes. Since these same processes and genes are involved during carcinogenesis, it is not surprising that hypoxia is a key tumoral micro-ambient factor involved in cancer progression.

### 2.2. The Hypoxia Signaling Pathway and Breast Cancer

Most solid tumors have a hypoxic environment, which correlates with poor clinical outcomes. Early reports by Hockey et al. demonstrated that low oxygen in tumors was associated with increased metastasis and lower survival in patients with breast cancer [9]. Indeed, it has been estimated that 40% of all breast tumors and 50% of locally advanced breast cancers have hypoxic regions at the time of diagnosis [10], which add to the role of hypoxia during the early tumor progression. In common with other tumors, breast cancer tissues present higher levels of HIF-1α and hypoxia, which correlates with poor prognosis, including early relapse and metastatic disease [11]. As expected by the oxygen diffusion ability, breast precursor lesions such as ductal carcinoma in situ (DCIS) and early stage breast cancer already present HIF-1α overexpression [12].

The HSP is key to two main cancer progression processes: angiogenesis and metabolic reprogramming. As stated before, oxygen diffusion is limited to 180 µm, so tumoral growth is restricted to masses not larger than 1–2 mm before becoming hypoxic [13]. After reaching that volume, the HSP induces a complex stress response aimed mainly at supporting neo-angiogenesis and metabolic reprogramming [14]. This process is not straight-forward, since most of the new vessels are disorganized and leaky, which further increases hypoxic areas. Neo-angiogenesis relies heavily in the HSP, mediated by classical angiogenic inducers such as VEGF and Angiopoietin-like factors and angiogenesis receptors (e.g., VEGFR, ANGPTR) and microenvironment matrix elements that act not only in the tumor cells themselves, but also in the tumor endothelial cells [15]. In breast cancer, several reports [16,17,18] have shown that a complex interplay between tumor and stromal cells create a pro-angiogenic environment in which the HSP mediated by HIF members is the key regulator, as shown by loss-of-function experiments [19].

The second important oncogenic process regulated by the HSP is metabolic reprogramming, which includes carbohydrates, amino acids, and lipids. The main example is the modulation of the cellular energetic metabolism by hypoxia. In this case, HIF-1α induces a shift from mitochondrial respiration to glycolytic-dependent metabolism. This is achieved through up-regulation of glycolytic enzymes and pyruvate redistribution toward lactate production by several coupled mechanisms [20]. In parallel, hypoxia induces a concurrent increase in production and secretion of lactate, which acidifies the extracellular milieu [21]. Increases in glycogen synthesis and glucose uptake also accompanied this change, which add to a chemoresistance phenotype of breast cancer cells [22]. Accumulation of extracellular lactate and consequent acidification contribute to an important immunosuppressive microenvironment in breast tumors [23]. The principal actors in these changes are the HIF-responsive carbonic anhydrase 9 and monocarboxylate transporters (MCT) 1 and 4 [23,24]. The former catalyzes the conversion of CO_2_ and water to HCO_3-_ and H_+_, whereas the latter mediates the lactate and H_+_ efflux from breast cancer cells [21,25]. Since most of the pyruvate in cancer cells is redirected away from the Krebs cycle, hypoxic cells require additional sources of Krebs cycle intermediates, such as cysteine and glutamine. This is achieved by an up regulation of several amino acid importers, such as SNAT2, SLC1A5, ASCT2, SLC7A11 and SLC7A5, all of which are HIF-responsive genes [26] and by an increase in the enzyme glutaminase, which converts glutamine to glutamate [27]. Cancer cells require fatty acids and lipids to support key ongoing oncogenic processes such as metabolism, signaling, intracellular oxidative adaptation and growth [28]. HIF proteins represses fatty acid oxidation and up regulates their synthesis by transactivating the genes of several enzymes involved in this process, such as the fatty acid synthase, lipin 1 and acetyl-CoA carboxylase (ACC). Simultaneously, the HSP is involved in an important increase in fatty acid uptake by up regulating fatty acid-binding proteins [29].

More recently, several authors have found that the HSP can modulate not only transient transcriptional responses, but also epigenetic programs. This is accomplished by changing the methylation status of both DNA and nuclear histones [30]. Induction of histone lysine demethylases (KDM) by the HSP is key to this process, as they stabilize HIF-1α complexes to initiate transcription of its key target genes. In addition, several KDM are members of the 2-oxoglutarate-dependent dioxygenase family (KDM3A, KDM2B, KDM4B, KDM5B, KDM6B and KDM4C), so they depend on oxygen and 2-oxoglutarate to present their enzymatic activity [30]. These enzymes act also as signal amplifiers and transcriptional facilitators for the expression of genes downstream of HIF signaling [31] and also mediate chromatin rearrangements to facilitate this [32]. Similarly, hypoxia reduces the activity of the TET demethylases, inducing DNA hyper-methylation [33]. Pediatric ependymomas underline the importance of hypoxia-mediated epigenetic reprogramming in cancer. As many of other pediatric tumors, these cancers have a very low number of recurrent mutations, so an epigenetic origin has been suggested. Ependymomas arise from putative stem cells embedded in a hypoxic compartment, in which the lack of oxygen establishes a gene regulatory program that depends on it [34,35]. These results show the HSP ability to regulate the epigenetic machinery to provide a stable response to hypoxia.

Several research groups have shown that a large part of what previously was considered “junk” DNA is actually transcribed [36] and at least part of it has important cellular functions [37,38]. Among the important transcribed DNA regions, non-coding RNAs genes have proven to be an enormous source of previously unknown regulatory elements that take part in physiological and pathological states. Among the latter, cancer stands out, perhaps due to the complex and profound genomic rearrangements that characterize it [39]. We can arbitrarily separate non-coding RNAs into two groups, small non-coding RNAs (sncRNAs) and long non-coding RNAs (lncRNAs), according to their length. Long non-coding RNAs (lncRNAs) are transcripts longer than 200 base pairs transcribed from intergenic or even genic regions. They are classified by its molecular function as decoy lncRNAs, when they sequester proteins, guide lncRNAs, which can recruit chromatin modifiers, scaffolding lncRNAs that act as protein adaptors and sponges, which act as competing endogenous molecules (ceRNAs) that prevent microRNAs to interact with mRNAs by being “sponges” and enhancer lncRNAs that stabilize chromosome loops [40]. These molecular activities allow them to take part in almost all cellular processes explored to date, as they converge into transcriptional and postranscriptional mechanisms, epigenetic modulation and even signal transduction participation. As expected by these facts, there are many reports showing deregulation of several lncRNAs in a long list of tumors [41]. Changes in lncRNAs expression have been also associated with clinical characteristics, prompting several authors to propose the use of them as diagnostic or prognostic tools [42]. Although the study of these RNAs has escalated in recent years, we are still in the initial steps of their understanding, due to the large number of lncRNAs, which may even exceed the number of coding RNAs [43], and their pleiotropic activity.

In breast cancer, lncRNAs have been extensively studied, but their role seems to be more complex that initially thought. These molecules participate in breast cancer cell proliferation [44,45], invasion [46], migration [45,46,47], apoptosis [48], epithelial-mesenchymal transition (EMT) [49], stem phenotype [50] and response to chemotherapeutic drugs [40,51] and most of the described cancer hallmarks [52]. Most of the studies have focused on the cell-autonomous effects of these RNAs, leaving the role of non-cell autonomous signaling provided by the Tumor Microenvironment (TME) unexplored. Since there is a remarkable intra- and inter-tumoral heterogeneity in breast cancer, more efforts are needed to study the role of the surrounding TME in gene expression regulation. In addition, recent reports have also shown that these non-coding RNAs can also take part in emerging cancer hallmarks such as phenotypic plasticity and non-mutational epigenetic reprogramming (NMER) [53]. TME is thus a key component required for full cancer progression, as it not only provides an adequate niche for tumor development, but also interacts with cancer cells in a bidirectional manner [54].

## 3. Hypoxia-Associated Long Non-Coding RNAs as Regulators of Breast Cancer

As stated previously, lncRNAs are critically involved in a bidirectional signaling circuit between TME and cancer cells. Two TME-related factors are key to drive cancer progression by providing important evolutionary forces involved in it: hypoxia and acidosis. In this review, we will focus only on the hypoxic response. In breast cancer, most of the attention has focused on the HIF-dependent or independent HSP, but new research has shown that non-coding RNAs may be key to understand the full response of tumors to hypoxia.

Recently, Lin et al. reported the discovery of hypoxia-regulated lncRNAs in breast cancer cells [55]. These authors used a RNASeq approach in MCF-7 cells exposed to normoxia, hypoxia and re-oxygenation conditions and found 472 lncRNAs that were differentially regulated during hypoxic conditions, validating three of them: lnc-CPN2-1, lnc-C11orf35-2 and lnc-NDRG1-1. Although not comprehensive since it was produced using only one cell line, this report showed that the number of hypoxia-responsive lncRNAs could be counted into the hundreds. A second article published in 2021 used two breast cancer cell lines to perform a similar RNASeq-based analysis [56]. Here, 104 and 282 regulated lncRNAs were found in each cell line, and only 43 were shared between them. Twenty-six RNAs were validated, but interestingly, 17 were not. Results from this study could help to provide more accurate network analyses and prognostic markers. For example, Gong, P. J. et al. recently used a panel of 13 hypoxia-related lncRNAs to generate a classification or “clusters” of breast cancer patients based on this signature [57]. The cluster with an unregulated hypoxic signature presented a differential immune infiltration profile, with lower CD8+ and CD4+ T but higher nTreg and iTreg cells number. A more comprehensive study using several cell lines and a more robust validation strategy, including in vivo experiments, is needed to establish the real number of hypoxia-related lncRNAs in breast cancer or at least those that are more important for the malignant phenotype. In addition, a study like this should help to add new diagnostic and prognostic factors and even possible therapeutic targets.

Most of the reports that have analyzed the role of lncRNA in the HSP have focused on RNAs that respond to hypoxia, although there are also several reports that have shown a direct role of lncRNAs that regulate the HSP and are even able to establish a positive signaling feedback to amplify the HSP [58,59]. Most of these reports analyzed lncRNAs that were previously associated with hypoxia in other tissues, so a more comprehensive discovery effort, as stated, should help to direct the efforts to the more relevant RNAs (Table 1).

The cellular processes modulated by hypoxic-associated lncRNAs are diverse, but most authors focused on proliferation, migration and invasion, epithelial-mesenchymal transition (EMT) and glycolysis, as these are probably the most studied hallmarks of cancer [52] (Table 1). Additional processes, such as apoptosis, stemness, and angiogenesis, were less studied, whereas recently described cancer hallmarks, such as non-mutational epigenetic reprogramming or senescence, have not been explored at all. We will next describe the role of specific hypoxia-related lncRNAs in each of these processes.

### 3.1. Growth

In 2015, Choudry et al. showed that hypoxia in breast cancer cells induced the formation of nuclear paraspeckles (a nuclear structure responsible for adenosine-to-inosine edition, among other functions). This led to the nuclear retention of the F11R transcript, due to the induction of the lncRNA NEAT1, which is part of this subcellular structure. Paraspeckle formation was associated with an increase in proliferation, enhanced clonogenic survival and reduced apoptosis [60]. Interestingly, the induction of NEAT1 depended on HIF-2α, not HIF-1α. More recently, three articles have shown that MALAT1, discovered initially in lung adenocarcinomas increase proliferation and invasion of breast cancer cells. Several mechanisms mediate this effect, including XBP1-HIF1α signaling [44], chromatin remodeling [65] and a ceRNA-mediated process that “sponge” the microRNA miR-3064-5p [66]. It is interesting to note that both HIF-1α and HIF-2α upregulate MALAT1 [66]. This lncRNA is regulated by hypoxia in several human cancers, but the main mechanism of action may vary between tumor types, as exemplified by the report showing that cytosolic functions of MALAT1 are more important in breast cancer [66], than in other tumors [84,85]. In 2020, the group of Cheng et al. demonstrated that the non-coding RNA LINC00662 increased the proliferation of breast cancer cells by acting as a sponge on miR-497-5p [68]. This action decreased the expression of the egl-9 family hypoxia-inducible factor 2 (EglN2), a prolyl hydroxylase that takes part in the hypoxia response. BRCT1 is a lncRNA that also regulates the proliferation of breast cancer cells by targeting miR-1303 [71]. The decrease of this miRNA prevents the degradation of PTBP3 mRNA, which is a breast tumor promoter. In addition to a role downstream of the HSP, there are several examples of long non-coding RNAs acting upstream of this signaling cascade (Table 1). VCAN-AS1 acts as a competitive endogenous molecule of miR-106a-5p [77]. The decrease in available miR-106a-5p stabilizes STAT3 mRNA and activates HIF-1α, inducing proliferation in breast cancer cells. Another example is HCG18, a lncRNA that induces proliferation of breast cancer cells by sponging miR-103a-3p, which is a regulator of the ubiquitin-conjugating enzyme W2O (UBE2O) [58]. This enzyme is part of the UBE2O/AMPKα2/mTORC1 signaling axis, which is also an inducer of the HCG18 by promoting the activation of HIF-1α and providing a positive feedback loop. Another example is LINC00649, which increased the stability and thus the levels of HIF-1α mRNA by interacting with the nuclear factor 90 (NF90)/NF45 complex, which is a signaling cascade responsive to double-stranded DNA, inducing proliferation of breast cancer cells [83]. GHET1 is a novel lncRNA that regulates the proliferation of breast cancer cells by decreasing HIF-1α expression by increasing the phosphorylation of LATS2 and YAP in order to inhibit the activation of the developmental Hippo pathway [79]. A similar example of lncRNA-mediated regulation of an important development-associated signaling pathway is mediated by RBM5-AS1, which is a lncRNA that is over expressed in breast cancer. This lncRNA increased proliferation after being induced by the RUNX2 transcription factor, which is also regulated by hypoxia [82]. This increase activated the Wnt pathway by preventing beta catenin degradation and stabilizing beta catenin-TCF4 transcriptional complexes. Interestingly, there are also examples of lncRNAs with a suppressor phenotype. For example, the novel hypoxia-induced mitochondrial-encoded RNA, mTORT1, reduced the proliferation of breast cancer cells by acting as a sponge for miR-26a-5p, which in turn regulated CREB1 and STK4 expression [74]. It is important to note that only two reports have shown that lncRNAs regulate tumor growth in vivo [56,59], so an additional effort is required to validate the role of these molecules on cell proliferation.

### 3.2. Migration, Invasion and Metastasis

Most of the reports that studied the role of hypoxia-associated lncRNAs in breast cancer proliferation also analyzed the effects of these molecules on migration, invasion, and metastasis [58,60,66,71,74,75,77,79,82,83,86,87,88]. Although this is an apparent bias caused by the researchers’ selection of analyzed phenotypes, there is indeed a biological basis for the shared cellular phenotype elicited by the lncRNAs, since most of the signaling cascades analyzed are oncogenic, which tend to have pleiotropic network effects. A clear example of this is the previously mentioned report the activation of the Wnt pathway by RBM5-AS1 [82], which induces not only proliferation but also migration, invasion, epithelial-mesenchymal transition (EMT) and stemness, all of which are cellular behaviors expected from the reported role of Wnt on development and carcinogenesis [89]. Nevertheless, there are three reports in the literature that analyzed only the role of hypoxia-related lncRNAs on migration, invasion, or metastasis.

The first of them is the analysis of the role of long non-coding RNAs expressed from the EFNA3 (Ephrin-A3) locus after hypoxia on the metastatic phenotype of breast cancer cells [61]. This effect was mediated by the increase of EFNA3 protein expression by an unknown mechanism. Since EFNA3 is a key cell surface protein required for efficient migration, invasion and metastasis of cancer cells, the induction of these lncRNAs could be one of the mechanisms associated with the effects of hypoxia on metastasis [90]. The second report involves the description of the effects of the well-known HOTAIR lncRNA. HOTAIR is highly expressed in breast cancer metastasis, inducing the overexpression of the HER2 oncogene through sequestration of miR-331-3p [91]. This lncRNA is induced by hypoxia in breast cancer cells where it acted as a competitive endogenous molecule toward miR-204, stimulating FAK1 mRNA degradation [72]. The third report showed that the intronic-derived SPRY4-IT1 increased breast cancer cell metastasis [80]. This lncRNA was shown to bind to the transcription elongation factor B subunit 1 (TCEB1) mRNA, increasing its stability, which in turn increased HIF-1α expression. Nevertheless, the molecular mechanism uncovered by these authors was described in colon cancer cells, so further corroboration of breast tumors is needed.

### 3.3. Glycolysis

Glycolysis is also one of the most studied metabolic pathways, so there are several reports of its modulation by hypoxia-related lncRNAs in breast cancer. The first analysis was reported by Lin et al., who found that LINK-A (long intergenic non-coding RNA for kinase activation) promoted the phosphorylation of HIF-1α by BRK and LRRK2 kinases [62]. This postraductional modification activated HIF in normoxic conditions, inducing glycolysis in breast cancer cells. Pyruvate dehydrogenase kinase 1 (PDK1) is a key enzyme in glucose metabolism. It has been shown that hypoxia induced this kinase by a mechanism that required the expression of the lncRNA H19 [64]. Furthermore, inhibition of this RNA decreased hypoxia-induced glycolysis. A similar postraductional mechanism was found for phosphoglycerate kinase, a key glycolytic enzyme. Enhanced ubiquitination by the E3 ligase STUB1 mediated by LINC00926 downregulated this enzyme [76]. MIR210HG is a hypoxia-induced lncRNA that promoted the Warburg effect in breast cancer cells [69]. This was achieved by increasing the translation of HIF-1α and the consequent transactivation of glycolytic enzymes. Finally, it is worth mentioning that GHET1 lncRNA not only increase proliferation of breast cancer cells, but also increases glycolysis in them [79].

### 3.4. Stemness

Tumor Initiating Cells (TIC) or Cancer Stem Cells (CSC) are a small subpopulation of tumoral cells that resemble progenitor tissue cells from the original organ from which the tumor arose. CSC are important drivers of initiation, progression, invasion, metastasis and drug resistance in all the tumors examined so far [92], including breast cancer [93]. Breast cancer stem cells (BCSC) were first isolated by Al-Hajj et al. using cell surface markers (EpCAM+/CD44+/CD24−) [94]. Abnormal expression of lncRNAs contribute to almost all aspects of cancer progression, as stated previously. In breast cancer, modulation of several lncRNAs contribute to the stemness phenotype, including well-known non-coding RNAs such as MALAT-1, HOTAIR and H19 [95]. Nevertheless, there is a paucity of reports exploring the role of these RNAs in the HSP. Previous reports have shown that inhibition of HIF-1α decreased the proportion of breast cancer stem cells in xenografts and that a hypoxic state induced more malignant traits in these cells [96,97]. As expected by these results, we [70] and other researchers [58,82] (Table 1) have shown that lncRNAs could be key unexplored regulators of the stem phenotype, by means of acting as ceRNAs for stem-related microRNAs, such as miR-103 [58,82,98]. Using a three-dimensional in vitro tumoral model we showed that the expression of lncMat2B, a lncRNA expressed in the hypoxic regions of multicellular tumor spheroids (MCTS), is increased in the CD44+/CD24- breast cancer cell subpopulation [70]. Inhibition of this lncRNA decreased the stem properties of this subpopulation, as assessed by clonogenic assays and in vivo xenografts. In a similar work, Liu et al. showed that a decrease in the expression of the lncRNA HCG18 reduced the expression of stem markers and the growth of tumors in a mice xenograft model [58]. Although not fully explored, preliminary reports support the view that microRNA sponging may activate stem signaling pathways or genes such as c-Myc [81] or AMPKα2 [58]. More research is clearly needed to address the full relevance of non-coding RNAs in breast cancer stem cells functions.

## 4. Hypoxia-Associated lncRNAs as Breast Cancer Clinical Molecular Markers

As expected by the role of hypoxia in cancer, several research groups have explored the potential use of hypoxia-related lncRNAs as molecular markers. In 2017, Liu et al., created a 3-lncRNA signature that could classify breast tumors as triple-negative or non-triple negative tumors based on plasma obtained from patients [99] (Table 2). Gong et al. used a 13-gene signature that included four lncRNAs for predicting poor prognosis of breast cancer patients using a network approach that also uncovered immunological differences [57]. A more recent work using a four-lncRNA signature had prognostic power for overall survival in patients with breast cancer when used to stratify patients in low- and high-risk groups [100]. Gu et al. recently proposed a more extensive signature with 12 lncRNAs [101]. This signature was used to predict the survival outcome, classifying breast cancer patients in high and low-risk groups. Patients in the high-risk group had shorter median overall and disease-free survival and lower chemosensitivity compared with those in the low-risk group. More importantly, the score provided by these authors was an independent prognostic factor. Finally, there are additional works that have explored the use of single lncRNAs as markers, such as the report from Wang et al. in 2019, which found that HIF1A-AS2 expression had prognostic value for lymph node and distant metastasis, unfavorable histological grade and shorter overall survival [102].

These recent reports are encouraging, but there is still the need to provide further validation in distinct populations using larger number of patients in studies that also incorporate breast cancer molecular subtypes in order to implement these as routine markers.

## 5. Discussion

Since the discovery that most of the human DNA is transcribed and have a potential function [36,37,38], extensive research has shown that non-coding RNAs genes provide a large number of regulatory elements in cancer. Due to the sheer number of these molecules and lower evolutionary conservation compared to coding DNA, we have only a fragmentary glimpse of their function in these diseases. Besides, the non-cell autonomous role of lncRNAs and the importance of TME in breast cancer progression make the task more difficult. Nevertheless, several advances have been made. In this review, we provided a general overview of the lncRNAs´ role in the hypoxic response in these tumors. Most of the studies provide information about single lncRNAs functions over the classic cancer hallmarks, and only a few of them aimed to discover all the possible hypoxic-responsive lncRNAs or the role of non-coding RNAs upstream of the HSP. High-throughput experiments in a larger number of cells lines stratified by breast cancer molecular subtypes with extensive in vivo validation are now required to gain a more comprehensive insight on the lncRNAs role in the hypoxic response. These results could not only be useful for this, but also to provide better diagnostic, prognostic, and perhaps therapeutic tools for breast cancer patients.

## Figures and Tables

**Figure 1 cells-11-01679-f001:**
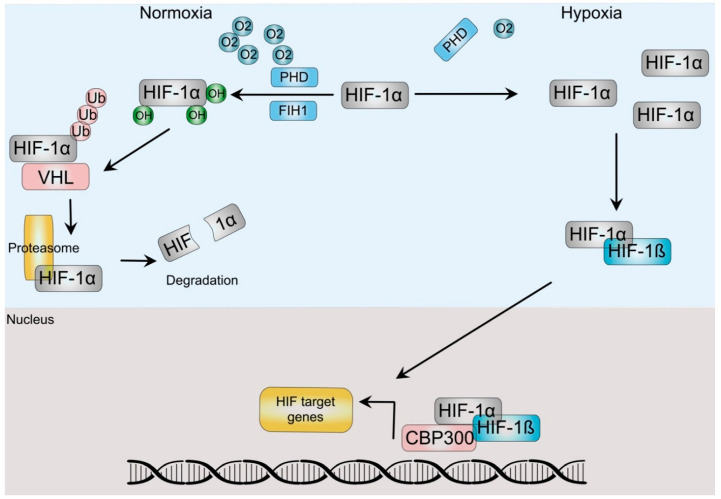
HIF-dependent hypoxia signaling pathway. In normoxia, prolyl-hydrolyses (PHD) and Factor-inhibiting hypoxia-inducible factor (FIH) hydroxylate HIF-α. Hydroxylation from the latter impairs HIF transactivation of target genes, whereas the former allows dimerization with the Von Hippel Lindau (VHL) protein, which directs HIF to ubiquitination and degradation by the proteasome. Hypoxic conditions stabilize HIF-α after PHD and FIH inactivation, allowing dimerization with HIF-ß subunits, translocation to the nucleus and association with coactivators, such as p300/CBP to regulate gene expression.

**Table 1 cells-11-01679-t001:** Hypoxia-related lncRNA in breast cancer.

Year	lncRNA	Expression in Breast Cancer	Regulated by Hypoxia	Regulates Hypoxia Signaling	Molecular	Phenotype	References
2015	NEAT1				Paraspeckle formation	Proliferation, apoptosis inhibition	[60]
2015	EFNA3				Induce Ephrin-A3 accumulation	Increased metastatic potential	[61]
2016	LINK-A	Increased in TNBC			Allows HIF-1α phosphorylation and stabilization by BRK and LRRK2	Glycolysis	[62]
2018	NDRG-OT1				Inhibition of NDRG1 by ubiquitination-mediated proteolysis. Inhibition of NDRG1 promoter activation		[55,63]
2018	H19				miRNA let-7 endogenous competitor. HIF-1α activator	PDK-1-mediated Increased glycolysis	[64]
2018, 2019, 2021	MALAT1, TALAM	Increased			XBP1-HIF-1α and HER2 pathway-mediated effects. miRNA-3064-5p endogenous competitor. Chromatin remodeling	Proliferation and invasion	[44,65,66]
2019	HISLA	Increased in breast cancer TAMs			HIF-1α stabilization by blocking PHD2	Macrophage-mediated enhanced glycolysis	[67]
2020	LINC00662	Increased			miR-497-5p endogenous competitor mediated EglN2 regulation	Proliferation and migration	[68]
2020	MIR210HG	Increased			HIF-1α translation enhancement	Glycolysis	[69]
2020	lncMat2B					Stemness	[70]
2020	BRCT1	Increased			miR-1303 endogenous competitor-mediated stabilization of PTBP3	Proliferation, migration, increased metastatic potential	[71]
2020	HOTAIR	Increased			miR-204 endogenous competitor-mediated FAK induction	Migration	[72]
2020	RAB11B-AS1	Increased			Induces the expression of VEGFA and ANGPTL4 by recruitment of RNA polymerase II	Migration, invasion, and angiogenesis	[73]
2021	LncIHAT				Promotes expression of PDK1 and ITGA6	Tumor growth and metastasis	[56]
2021	MTORT1				miR-26a-5p endogenous competitor-mediated CREB1 and STK4 regulation	Proliferation and migration	[74]
2021	UCA1	Increased				Proliferation and apoptosis	[75]
2021	LINC00926				Regulation of PGK1 via ubiquitination mediated by STUB1	Glycolysis	[76]
2021	VCAN-AS1	Increased			miR-106a-5p endogenous competitor-mediated regulation of STAT3	Proliferation, migration, invasion, and EMT.	[77]
2021	HCG18	Increased			miR-103a-3p endogenous competitor-mediated UBE2O/AMPKα2/mTORC1 activation	Proliferation, invasion and stemness	[58]
2021	eNEMAL				Alternative splicing of NEAT1		[78]
2021	PCAT-1	Increased			Stabilizes HIF-1α by preventing RACK1 binding		[46]
2021	GHET1	Increased in TNBC			Decreased phosphorylation of LATS1 allowing YAP nuclear translocation	Proliferation, viability, invasion, glycolysis	[79]
2021	SPRY4-IT1				STAU1 recruitment to TCEB1, upregulating HIF-1α	Metastasis	[80]
2021	HIFAL	Increased			Recruitment of PHD3 to PKM2 to enhance HIF-1α activity	Tumor growth	[59]
2021	KB-1980E6.3	Increased			Activation of IGF2BP1/c-Myc signaling axis to stabilize c-Myc	Stemness	[81]
2022	RBM5-AS1	Increased			Activation of the Wnt pathway	Proliferation, migration, invasion, EMT, and stemness	[82]
2022	LINC00649	Increased in TNBC			Stabilization of HIF-1α via NF90/NF45 interaction	Proliferation, migration, and invasion	[83]

TNBC: Triple Negative Breast Cancer; TAMs: Tumor-Associated Macrophages. Gray bars show regulation (upstream or downstream)

**Table 2 cells-11-01679-t002:** Hypoxia-associated molecular lncRNA markers.

Year	Marker Genes	Signature Derived From	Features	Reference
2017	3-gene signature. ANRIL, HIF1A-AS2, UCA1	Plasma	Differentiate TNBC vs. other	[99]
2020	13-gene signature that includes SNHG16, LINC00899, PSMG3-AS1 and PAXIP-AS1 lncRNAs	Tumoral tissue	Prognosis	[57]
2021	12-gene signature. LINC01614, LINCO2384, AL109955.1, AC044849.1, LINCO2084, LINC01615, YTHDF3-AS1, AL451085.3, AL512380.1, HSD11B1-AS1, AC110995.1, AC004847.1, TRG-AS1, AC011978.2, CDK6-AS1, TDRKH-AS1, OTUD6B-AS1, MIR4435-2HG, AP003774.2, SLC12A5-AS1	Tumoral tissue	Prognosis and immune features	[101]
2021	4-gene signature. AL031316.1, AC004585.1, LINC01235, and ACTA2-AS1	Tumoral tissue	Prognosis	[100]
2018	MALAT1	Tumoral tissue	Number of metastatic lymph nodes	[44]
2019	HIF1A-AS2	Tumoral tissue	Lymph node metastasis, distant metastasis, and unfavorable histological grade. Shorter overall survival	[102]

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
