# Peer review of "The Role of Hypoxia-Associated Long Non-Coding RNAs in Breast Cancer"

_cells, 2022, doi:10.3390/cells11101679_

Round 1

Reviewer 1 Report

The manuscript entitled: “The role of hypoxia-associated long non-coding RNAs in breast  cancer”  is a well written manuscript in a field that is being investigated thoroughly. For this reason there are many reviews and original articles that contribute to the evaluation of lncRNAs in breast cancer.

Although the list of references provided in the manuscript is detailed and most of the publications are from recent years, I have three more references that, in my opinion, have to be added to support the involvement of lncRNAs in the metastasis of breast cancer. These are the following:

Mondal, P., & Meeran, S.M. (2020). Long non-coding RNAs in breast cancer metastasis. Non-coding RNA Research, 5, 208 - 218.

Liu, L., Zhang, Y., & Lu, J. (2020). The roles of long noncoding RNAs in breast cancer metastasis. Cell Death & Disease, 11.

Zhang, T., Hu, H., Yan, G., Wu, T., Liu, S., Chen, W., Ning, Y., & Lu, Z. (2019). Long Non-Coding RNA and Breast Cancer. Technology in Cancer Research & Treatment, 18.

I have also noticed some minor errors in the text:

Line 188, a connective word is missing between deregulation and several

Line 248, provide explanation of NMER abbreviation

Line 362, change “a” in high a low-risk groups

Line 371, a word is missing in between “order implement”

Author Response

Thank you for your comments. 
Although the list of references provided in the manuscript is detailed and most of the publications are from recent years, I have three more references that, in my opinion, have to be added to support the involvement of lncRNAs in the metastasis of breast cancer. These are the following:

Mondal, P., & Meeran, S.M. (2020). Long non-coding RNAs in breast cancer metastasis. Non-coding RNA Research, 5, 208 - 218.

Liu, L., Zhang, Y., & Lu, J. (2020). The roles of long noncoding RNAs in breast cancer metastasis. Cell Death & Disease, 11.

Zhang, T., Hu, H., Yan, G., Wu, T., Liu, S., Chen, W., Ning, Y., & Lu, Z. (2019). Long Non-Coding RNA and Breast Cancer. Technology in Cancer Research & Treatment, 18.

All the references were added to the text

I have also noticed some minor errors in the text:

Line 188, a connective word is missing between deregulation and several
Added

Line 248, provide explanation of NMER abbreviation
Eliminated the abbreviation 

Line 362, change “a” in high a low-risk groups
Changed

Line 371, a word is missing in between “order implement”
Added the missing word

Reviewer 2 Report

In this review, Maldonado and Melendez-Zajgla describe the role of some long non-coding RNAs (lncRNAs) associated to hypoxia in breast cancer development. The study of the participation of lncRNAs associated to hypoxia in breast cancer is novel; however there is a recent review that focuses on the role of hypoxia-induced lncRNAs in cancer (https://doi.org/10.3390/ijms22041857); thus authors should revise this previous review in order to avoid overlapping with some of the subjects that both reviews discuss (see Hypoxia Signaling Pathway). About 40% of the review is dedicated to review generalities of hypoxia-related pathways, which had been covered in several previous reviews. This part should be shortened to focus on the role of lncRNAs. “Stemness”, for example, is laconically described and the lncRNAs promoting this phenomenon are not mentioned. With some exceptions, authors did not explain whether the effect of hypoxia-associated lncRNAs on breast cancer cells is associated to a molecular phenotype in particular, is there any information in this regard? This is of particular relevance in the subject of “… lncRNAs as molecular markers”, the authors did not point out whether the mentioned studies make this distinction. Also it would help if authors clearly indicate whether the studies used lncRNAs obtained from plasma, biopsies or other type of sample, because this is only indicated in the first study they discussed.

There are some definitions or terms that must be corrected. For instance, glycolysis is not a cellular phenotype but a metabolic pathway (line 328, page 8); the term “junk DNA transcription” (line 374, page 10); “genes … regulate various biological process ” (line 57, page 2), where authors must be referring to gene products.

It would help the reader to indicate whether the lncRNAs shown in Table 1 are either up- or down-regulated. The role of hypoxia, or why authors focus on this phenomenon should be incorporated in the Abstract. 

Author Response

Thank you for your comments. Please find below our response.

In this review, Maldonado and Melendez-Zajgla describe the role of some long non-coding RNAs (lncRNAs) associated to hypoxia in breast cancer development. The study of the participation of lncRNAs associated to hypoxia in breast cancer is novel; however there is a recent review that focuses on the role of hypoxia-induced lncRNAs in cancer (https://doi.org/10.3390/ijms22041857); thus authors should revise this previous review in order to avoid overlapping with some of the subjects that both reviews discuss (see Hypoxia Signaling Pathway). About 40% of the review is dedicated to review generalities of hypoxia-related pathways, which had been covered in several previous reviews. This part should be shortened to focus on the role of lncRNAs.    We shortened and adapted this section, as suggested.   “Stemness”, for example, is laconically described and the lncRNAs promoting this phenomenon are not mentioned.    There is a lack of articles linking stemness and hypoxia in breast cancer. As the review focused on the latter, we omitted the information for the general role of stemness in breast cancer. Nevertheless, we improved the description of stemness in breast cancer and added more information.     With some exceptions, authors did not explain whether the effect of hypoxia-associated lncRNAs on breast cancer cells is associated to a molecular phenotype in particular, is there any information in this regard?    All the molecular information available was described. Several of the studies just reported the cellular effect of the specific lncRNA, without delving into more mechanistic aspects. This is indeed an area that needs more attention. As to the association of the lncRNA function to a specific breast cancer molecular subtype, there were only three papers that analyzed an specific subgroup (Triple Negative Breast Tumors). These were pointed out in a new added column. The rest of the articles used a combination of cell lines without trying to correct for molecular subtypes.   This is of particular relevance in the subject of “… lncRNAs as molecular markers”, the authors did not point out whether the mentioned studies make this distinction. Also it would help if authors clearly indicate whether the studies used lncRNAs obtained from plasma, biopsies or other type of sample, because this is only indicated in the first study they discussed.   We added the information and clarified that molecular markers refer to clinical molecular markers. All the papers used breast cancer samples without stratifying by molecular subtypes.   There are some definitions or terms that must be corrected. For instance, glycolysis is not a cellular phenotype but a metabolic pathway (line 328, page 8); the term “junk DNA transcription” (line 374, page 10); “genes … regulate various biological process ” (line 57, page 2), where authors must be referring to gene products.   Corrected   It would help the reader to indicate whether the lncRNAs shown in Table 1 are either up- or down-regulated.    We added a new column with the information   Please note that we eliminated reference 68 (1. Guo, X.; Lee, S.; Cao, P. The inhibitive effect of sh-HIF1A-AS2 on the proliferation, invasion, and pathological damage of breast cancer via targeting miR-548c-3p through regulating HIF-1α/VEGF pathway in vitro and vivo. Onco Targets Ther 2019, 12, 825-834, doi:10.2147/ott.S192377.) since the article has been recently retracted.

Reviewer 3 Report

Considering the volume of manuscript, it is recommended to focus only breast cancer with hypoxia pathway in introduction. I think it is not necessary to explain importance hypoxia pathway in cancers.  

Author Response

Thank you for your comments. Please find our response below

Considering the volume of manuscript, it is recommended to focus only breast cancer with hypoxia pathway in introduction. I think it is not necessary to explain importance hypoxia pathway in cancers.     We have deleted part of the hypoxia pathway section, to balance the article.    The role of hypoxia, or why authors focus on this phenomenon should be incorporated in the Abstract.    We corrected the abstract   Please note that we eliminated reference 68 (1. Guo, X.; Lee, S.; Cao, P. The inhibitive effect of sh-HIF1A-AS2 on the proliferation, invasion, and pathological damage of breast cancer via targeting miR-548c-3p through regulating HIF-1α/VEGF pathway in vitro and vivo. Onco Targets Ther 2019, 12, 825-834, doi:10.2147/ott.S192377.) since the article has been recently retracted.  

Round 2

Reviewer 3 Report

None